

# Comparative analysis of volume growth processes of Masson pine and Chinese fir forests in different regions of southern China

YuHui Chen and Zongzheng Chai

College of Forestry, Guizhou University, Guiyang, Guizhou, China

## ABSTRACT

Masson pine (*Pinus massoniana* Lamb.) and Chinese fir (*Cunninghamia lanceolata* (Lamb.) Hook.) are important coniferous species commonly found in southern China and play crucial ecological and economic roles. Understanding how regionally variable conditions influence their growth patterns can support effective forest management strategies and conservation efforts. Here, we used the Richards growth equation to comprehensively analyze their volumetric growth processes through multiple diverse regions in southern China, representing a pioneering large-scale application of unified modeling techniques. This work provides theoretical and technical knowledge to support the sustainable stewardship of these vital forest ecosystems. We found that: (1) The highest per-hectare volume accumulation occurs in southwestern China, at 97.455 $m^3$ $hm^{-2}$ for *P. massoniana* and 85.288 $m^3$ $hm^{-2}$ for *C. lanceolata*. These values are higher than in the southeastern (71.424 $m^3$ $hm^{-2}$ and 79.520 $m^3$ $hm^{-2}$, respectively) or south-central regions (70.697 $m^3$ $hm^{-2}$ and 65.647 $m^3$ $hm^{-2}$), predominantly due to varying stand maturity across these regions. (2) Both species exhibit age-dependent growth patterns in the first 100 years of forest development, characterized by rapid early growth and transitioning into slower, stable phases. The highest total volume growth for *P. massoniana* occurred in the southwest, followed by the southeast and south-central regions. Conversely, *C. lanceolata* growth is highest in the southeast, followed by south-central and the southwest. (3) Quantitative maturity for *P. massoniana* ranges from 27 to 29 years (average: 30 years) whereas *C. lanceolata* matures earlier, between 16 to 19 years (average: 20 years). Climate and soil factors significantly influence their growth dynamics, with precipitation, temperature variation, and soil characteristics driving differences in suitability and growth potential across distinct regions in southern China. Tailored management practices that align with local climatic and environmental conditions are recommended to optimize growth and ensure sustainable management and development of *P. massoniana* and *C. lanceolata* forests.

Corresponding author
Zongzheng Chai, chaizz@126.com

## INTRODUCTION

Masson pine (*Pinus massoniana* Lamb.) and Chinese fir (*Cunninghamia lanceolata* (Lamb.) Hook.) are coniferous species commonly found in southern China, known for their rapid growth and ecological, economic, and social significance. In addition to the important roles they play in their native ecosystems, these trees also strongly influence human development and well-being (*Niu et al., 2021*; *Ji et al., 2022*). Ecologically, *P. massoniana* and *C. lanceolata* have key functions in soil and water conservation, climate moderation, and biodiversity preservation. They provide habitats for wildlife and help sustain ecological equilibrium (*Gu et al., 2019*; *You, Zhu & Deng, 2021*; *Gong et al., 2023*). Economically, timber harvested from these species is widely used in construction, furniture making, and various other industries (*Siry, Cubbage & Ahmed, 2003*; *Qiao, 2014*). Moreover, *P. massoniana* and *C. lanceolata* are culturally significant, reflecting the unique natural characteristics and traditional practices of the regions they inhabit, and are intricately intertwined with human cultural heritage.

Comparative analysis of the volumetric growth characteristics of *P. massoniana* and *C. lanceolata* can enhance understanding of their ecological adaptability. From an ecological standpoint, better understanding of their growth is crucial for evaluating their functions within their respective ecosystems and can inform forest resource planning and management approaches. In forestry, such comparisons provide empirical evidence that can be used to guide species selection, afforestation strategies, and management practices. Variation in growth characteristics may affect timber yield and quality, thereby influencing economic returns. Furthermore, this work lays the groundwork for future research and enriches our understanding of tree growth dynamics. A systematic examination of developmental processes of *P. massoniana* and *C. lanceolata*, focusing on their distinct characteristics, offers valuable guidance for the use and sustainable development of forest resources.

While previous studies have explored the growth patterns of these species, most have either focused on localized analyses or did not use a standardized growth model across ecological regions they examined. This research addresses this gap by applying the Richards growth model to evaluate growth dynamics comprehensively across a diversity of regions within southern China, providing a uniform analytical framework that has not been previously applied at this scale for these species. The novelty of this study lies in its large-scale comparative approach, examining how regional differences in climate and soil impact growth patterns in *P. massoniana* and *C. lanceolata*. Unlike previous research, this study systematically analyzes volumetric growth across southwest, south-central, and southeast China, allowing for an unprecedented examination of regional variation in growth and its ecological implications. Such analysis is critical for developing targeted, region-specific management strategies that can enhance sustainable forest stewardship.

By filling these research gaps, our findings contribute to understanding of the ecological adaptability and management needs of these essential conifer species, improving forest management practices that support both productivity and ecological conservation.

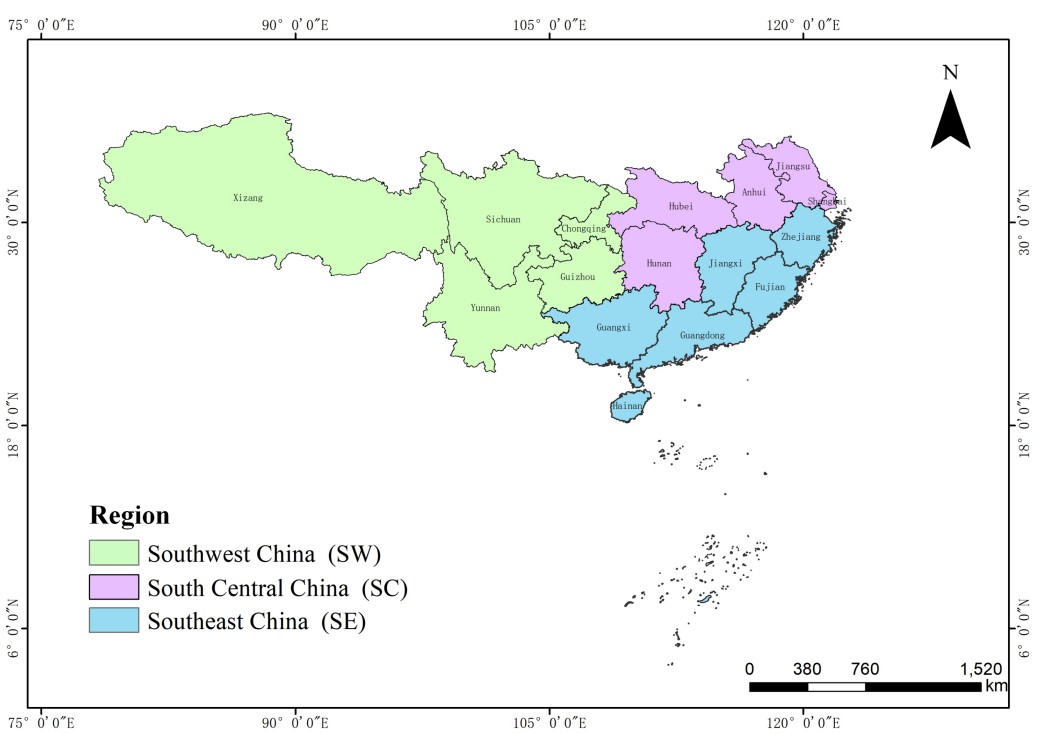

**Figure 1** **Regional divisions in southern China.** The southern region of China is divided into three areas: southwest China (SW), which includes Sichuan, Chongqing, Guizhou, Yunnan, and Tibet; south-central China (SC), encompassing Jiangsu, Shanghai, Anhui, Hubei, and Hunan; and southeast China (SE), consisting of Zhejiang, Jiangxi, Fujian, Guangdong, Guangxi, and Hainan.

## MATERIALS AND METHODS

### Geographic divisions in Southern China

This study focuses on the southern expanse of China, a region with distinctive climatic and natural geographical features. We divided this area into three provinces: southwest China (SW), encompassing Sichuan, Chongqing, Guizhou, Yunnan, and Tibet; south-central China (SC), which includes Jiangsu, Shanghai, Anhui, Hubei, and Hunan; and southeast China (SE), comprising Zhejiang, Jiangxi, Fujian, Guangdong, Guangxi, and Hainan (Fig. 1).

### Regional growth models

This study applied the Richards growth equation to analyze the relationship between volume and age for dominant tree species in each of the studied regions. This model was chosen for its strong applicability and accuracy, as demonstrated with National Forest Inventory data. *Fu, Zhang & Wang (2022)* were the first to apply this model on a national scale across thousands of sample plots (*i.e.,* 2,242 plots for *P. massoniana* and 3,013 for *C. lanceolata*), which provided a valuable reference for selecting this model. Extensive sample data ensures model reliability in fitting growth, making it particularly suitable for analyzing forest growth processes across different regions. Most of the models' correlation coefficients exceeded 0.6, and model predictions were validated against survey data collected

**Table 1   The growth equation parameters (a, b, c) for Masson Pine and Chinese Fir species, along with the number of sample plots, survey time, and fitted R-squared values across various regions in southern China.**

| Tree species | Region | Sample plots | Survey time | a | b | c | R² |
|---|---|---|---|---|---|---|---|
| Masson Pine | Southwest China (SW) | 386 | 1994–2018 | 176.927 | 2.045 | 0.045 | 0.740 |
| | South Central China (SC) | 939 | 1994–2018 | 151.518 | 1.579 | 0.029 | 0.616 |
| | Southeast China (SE) | 917 | 1994–2018 | 142.338 | 1.752 | 0.039 | 0.641 |
| Chinese Fir | Southwest China (SW) | 483 | 1994–2018 | 123.337 | 3.352 | 0.140 | 0.759 |
| | South Central China (SC) | 1,007 | 1994–2018 | 149.544 | 1.786 | 0.056 | 0.693 |
| | Southeast China (SE) | 1,523 | 1994–2018 | 183.207 | 1.355 | 0.037 | 0.580 |

Notes.
The survey period spanned from 1994 to 2018, with a total of four surveys conducted at five-year intervals.

across multiple periods. This work represents a pioneering comparative study of volume growth dynamics using large, multi-period datasets and the Richards growth equation to formulate precise growth models for various tree species across distinct regional landscapes in China.

Adapting the approach used by *Fu, Zhang & Wang* for major dominant tree species across China, we modified the Richards growth equation to relate volume and age for *P. massoniana* and *C. lanceolata* trees in southern regions:

$$V = a(1 - e^{-ct})^b$$

Here, *V* represents hectare volume ($m^3 \, hm^{-2}$), *t* denotes stand age (years), and constants a, b, and c are parameters specific to the growth equations for *P. massoniana* and *C. lanceolata* (Table 1).

### Climate and terrain data

To elucidate the effect of interactions between climate and terrain on *P. massoniana* and *C. lanceolata* forests in southern China, climate and terrain datasets were downloaded from the Resource and Environmental Science Data Center (https://www.resdc.cn) and the National Earth System Science Data Center (https://www.geodata.cn). Extracted variables included annual average temperature, mean temperature of the warmest month (July), mean temperature of the coldest month (January), annual precipitation, elevation, and slope.

## RESULTS

### Forest distribution

According to data from the 9th National Forest Resources Inventory (*National Forestry and Grassland Administration, 2019*), China's forested area encompasses a total of 1,798,885 square kilometers, with an aggregate volume of 1,705,819,590 cubic meters. The statistical results of *P. massoniana* and *C. lanceolata* forest resources (Table 2) indicate that: *P. massoniana* forests occupy 80,430 square kilometers, accounting for 4.471% of the national forest area, with a volume of 6,260,628 cubic meters, or 3.670% of total forest volume. *C. lanceolata* forests cover 113,866 square kilometers, representing 6.329% of the

**Table 2  Area and volume of Masson pine and Chinese fir forests in different regions of southern China.**

| Region | Age group | Masson Pine | | | Chinese Fir | | |
|---|---|---|---|---|---|---|---|
| | | Area ($10^2$ hm$^2$) | Volume ($10^2$ m$^3$) | per-hectare volume (m$^3$ hm$^{-2}$) | Area ($10^2$ hm$^2$) | Volume ($10^2$ m$^3$) | Per-hectare volume (m$^3$ hm$^{-2}$) |
| Southwest China (SW) | Young | 4,156 | 154,380 | 37.146 | 8,123 | 229,534 | 28.257 |
| | Middle-aged | 6,552 | 608,249 | 92.834 | 6,291 | 602,594 | 95.787 |
| | Near-mature | 6,741 | 798,558 | 118.463 | 3,207 | 409,881 | 127.808 |
| | Mature | 3,221 | 443,735 | 137.763 | 3,724 | 506,014 | 135.879 |
| | Over-mature | 162 | 25,258 | 155.914 | 835 | 143,659 | 172.047 |
| | Subtotal | 20,832 | 2,030,180 | 97.455 | 22,180 | 1,891,682 | 85.288 |
| South Central China (SC) | Young | 9,206 | 341,204 | 37.063 | 25,723 | 877,206 | 34.102 |
| | Middle-aged | 14,091 | 1,051,673 | 74.634 | 12,139 | 990,152 | 81.568 |
| | Near-mature | 7,285 | 631,119 | 86.633 | 4,213 | 543,946 | 129.111 |
| | Mature | 3,458 | 371,815 | 107.523 | 3,818 | 515,372 | 134.985 |
| | Over-mature | 144 | 20,911 | 145.215 | 1,106 | 160,014 | 144.678 |
| | Subtotal | 34,184 | 2,416,722 | 70.697 | 46,988 | 3,084,599 | 65.647 |
| Southeast China (SE) | Young | 4,053 | 89,974 | 22.199 | 15,677 | 401,560 | 25.615 |
| | Middle-aged | 10,604 | 638,928 | 60.253 | 11,842 | 890,037 | 75.159 |
| | Near-mature | 6,324 | 585,646 | 92.607 | 5,991 | 709,581 | 118.441 |
| | Mature | 2,830 | 362,072 | 127.941 | 8,658 | 1,177,267 | 135.974 |
| | Over-mature | 228 | 40,333 | 176.899 | 1,911 | 326,734 | 170.975 |
| | Subtotal | 24,039 | 1,716,953 | 71.424 | 44,079 | 3,505,179 | 79.520 |
| Southern China (Total) | Young | 17,415 | 585,558 | 33.624 | 49,523 | 1,508,300 | 30.457 |
| | Middle-aged | 31,247 | 2,298,850 | 73.570 | 30,272 | 2,482,783 | 82.016 |
| | Near-mature | 20,350 | 2,015,323 | 99.033 | 13,411 | 1,663,408 | 124.033 |
| | Mature | 9,509 | 1,177,622 | 123.843 | 16,200 | 2,198,653 | 135.719 |
| | Over-mature | 534 | 86,502 | 161.989 | 3,852 | 630,407 | 163.657 |
| | Subtotal | 79,055 | 6,163,855 | 77.969 | 113,258 | 8,483,551 | 74.905 |

total forested area, with a volume of 8,520,165 cubic meters, or 4.994% of overall forest volume. *P. massoniana* and *C. lanceolata* forests are primarily found in SC, where they occupy 34,184 and 46,988 square kilometers, respectively. In SE, they cover 24,039 and 44,079 square kilometers, respectively. They are least abundant in SW, where they occupy 20,832 and 22,180 square kilometers, respectively. Volume per hectare is highest in SW, followed by SE, with relatively low per-hectare volumes in SC. This is largely due to the abundance of juvenile forests in SC, whereas SW and SE have larger proportions of nearly mature stands.

## Forest growth dynamics

The Richards growth equation was used to calculate fitted growth volumes for *P. massoniana* and *C. lanceolata* forests across southern China from 0 to 100 years (Fig. 2). This analysis revealed a pattern of rapid initial growth, followed by stabilization as the forests mature. For *P. massoniana*, total growth volume peaked in SW, followed by SE and CS. *C. lanceolata* growth volume was highest in SE, followed by SC, with relatively low growth volumes in

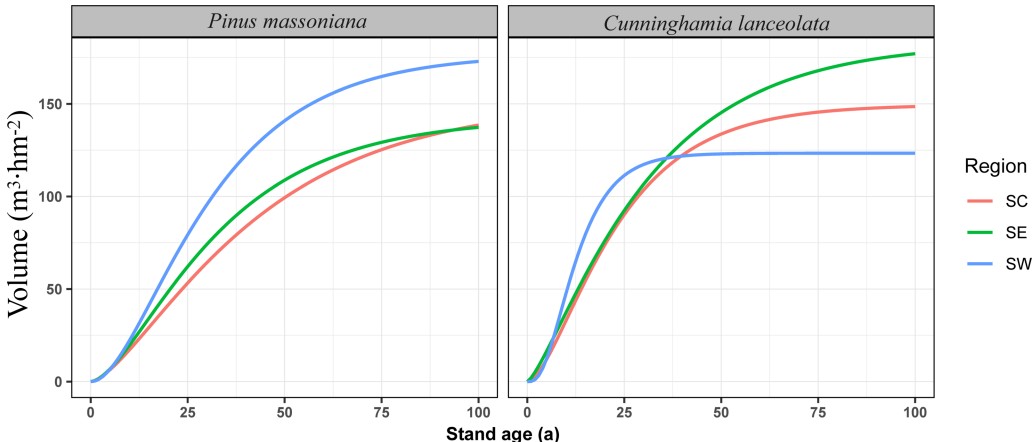

**Figure 2** **Volume growth dynamic for the first 100 years of Masson pine and Chinese fir forests in different regions of southern China.** The left panel shows the volume growth for *Pinus massoniana*, and the right panel shows the volume growth for *Cunninghamia lanceolata*. The regions are color-coded as follows: south-central China (SC) in red, southeastern China (SE) in green, and southwestern China (SW) in blue. The $y$-axis represents stand volume ($m^3$ hm), and the $x$-axis represents stand age (years). These curves highlight the regional differences in growth rates and volume accumulation for both species.

SW. Additionally, SW *C. lanceolata* forests had a phase of rapid early growth (0–35 years), with growth rates substantially exceeding those in SE and SC before decelerating and stabilizing after 35 years.

## Average and annual growth volumes

In SW, annual growth volume was higher than average growth volume in *P. massoniana* forests for the first 28 years when they both reached 3.2 $m^3$ $hm^{-2}$ $yr^{-1}$. This indicates that *P. massoniana* forests reach maturity at 28 years in this region. Average growth volume exceeded annual growth volume after this point. Conversely, annual growth volume was higher than average growth volume in *C. lanceolata* forests for the first 16 years, during which they were both 5.2 $m^3$ $hm^{-2}$ $yr^{-1}$. Following maturity, average growth volume exceeded annual growth volume. Notably, annual growth volumes were higher for *C. lanceolata* relative *P. massoniana* forests during the first 16 years, with a marked annual increase. However, annual growth volume declined progressively from years 10 to 40, after which they stabilized near zero. Moreover, average growth volume was higher for *C. lanceolata* relative to *P. massoniana* forests until year 38, after which it was lower for *C. lanceolata* forests.

In SC, annual growth volume was higher than average growth volume in *P. massoniana* forests until forest maturity at year 29, when they were equivalent at 2.1 $m^3$ $hm^{-2}$ $yr^{-1}$. Average growth volume was higher than annual growth volume after 29 years. For *C. lanceolata* forests, annual growth volume was higher than the average until the 19th year, when both reached 3.8 $m^3$ $hm^{-2}$ $yr^{-1}$, indicating forest maturity. Following maturity, average growth volume was higher than annual growth volume. Notably, the average growth volume of *C. lanceolat* a forests consistently exceeded that of *P. massoniana* forests.

*C. lanceolata* forests had higher annual growth volumes for the first 30 years, after which they began to decline more rapidly than annual growth volumes in *P. massoniana* forests.

In SE, annual growth volume was higher than average growth volume in *P. massoniana* forests until maturity at 27 years, when both rates were equivalent, at 2.5 m$^3$ hm$^{-2}$ yr$^{-1}$. Following maturity, average growth volume exceeded annual growth volume. In *C. lanceolata* forests, annual growth volume was higher than average growth volume until maturity at 16 years, when they both reached 3.9 m$^3$ hm$^{-2}$ yr$^{-1}$. Average growth volume was higher than annual growth volume after maturity. Overall, both annual and average growth volumes of *C. lanceolata* forests exceeded those of *P. massoniana* forests in SE.

Across southern China, maturation age ranged from 27 to 29 years for *P. massoniana* forests and was approximately 20 years for *C. lanceolata* forests (Fig. 3).

## Climatic and terrain variability across Southern China

From its coastal fringes to its inland territories, southern China exhibits pronounced climatic and topographical variation. SE coastal areas receive substantial annual rainfall, typically between 1,600 and 2,500 mm, with precipitation gradually diminishing with distance inland. In the SW, areas such as the Sichuan Basin and Chongqing have a mean annual precipitation of approximately 1,000 mm, reflecting a transition from humid to semi-arid conditions. Similarly, average annual temperature decreases with distance from maritime regions in the SE, such as Guangdong and Fujian, where maritime moderation results in temperatures between 19 and 23 °C. In contrast, temperatures in areas of the SW, such as Guizhou and Yunnan, average between 14 to 18 °C, with Tibet experiencing more extreme temperatures between 6 and 10 °C. Temperature also varies seasonally, with milder winter temperatures between 8 and 16 °C along the coastal SE, while inland areas in SW sometimes reach temperatures below −10 °C. In the summer, temperatures range from 27 to 30 °C along the coast, gradually decreasing with distance inland. Topographically, SE has lower-lying plains and hills, with elevation ranging from 500 to 1,500 m. In SW, which includes regions like Guizhou and Yunnan, elevation ranges from 1,000 to 3,000 m, with the Tibetan Plateau reaching over 4,000 m. Slopes become increasingly steep with distance inland, particularly in the rugged SW terrain (Fig. 4).

## DISCUSSION

Climate is one of the most important factors determining species distributions. *P. massoniana* and *C. lanceolata* prefer sunny and moist climates. The warm and humid conditions of southern China's subtropical climate are highly suitable for the growth of *P. massoniana* and *C. lanceolata* (*Zhang et al., 2020*; *Fu et al., 2017*; *Wang et al., 2022*). Additionally, both species have low soil requirements and are highly adaptable. Southern China is mountainous and is home to a diversity of soil types, including sandy, loamy, and acidic soil, all of which support the growth of both tree species (*Kuang et al., 2008*; *Mei et al., 2021*). Thus, the combination of climate and soil conditions makes southern China suitable for the growth of *P. massoniana* and *C. lanceolata* (*Jing et al., 2022*; *Wu, Duan & Zhang, 2019*).

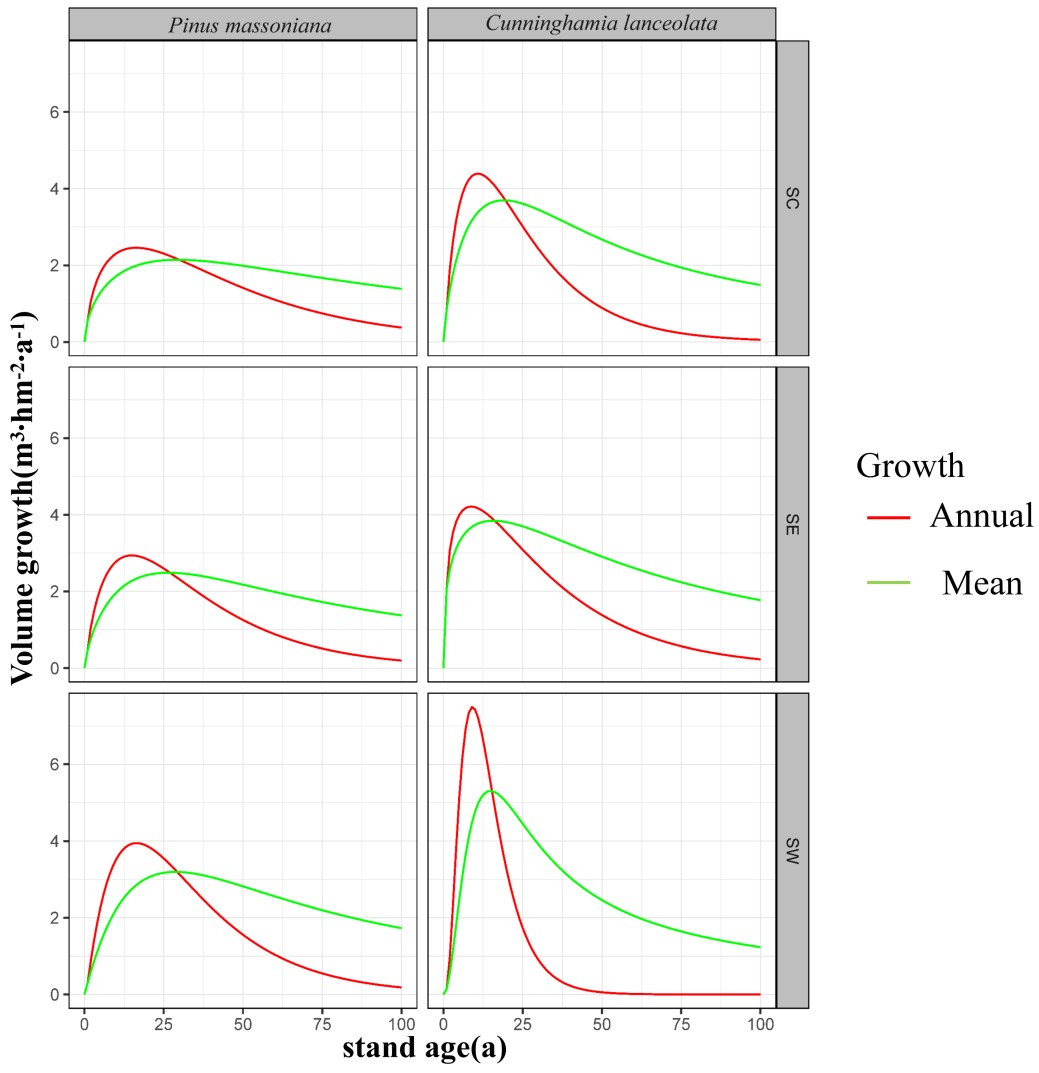

**Figure 3** **Volume growth curves for the first 100 years of Masson pine and Chinese fir forest growth in different regions of southern China.** The left column represents the growth of Pinus massoniana, while the right column represents Cunninghamia lanceolata. The regions are divided into south-central China (SC), southeastern China (SE), and southwestern China (SW) from top to bottom. The red lines indicate annual growth, and the green lines represent mean growth over time. The $y$-axis shows stand volume (m³ hm$^{-2}$), and the $x$-axis represents stand age (years). These graphs illustrate the variations in growth patterns between different regions and species.

Southern China extends from coastal areas inland, with precipitation gradually decreasing and site conditions deteriorating with the transition from SE to SW. Consequently, *C. lanceolata* becomes more narrowly distributed while the distribution of ponderosa pines expands. In terms of stand growth, *C. lanceolata* forests are largest in SE and smallest in SW, where growth slows with advancing stand age. This is primarily due to *C. lanceolata*'s preference for light, deep, loose and relatively fertile soil rich in humus (*Huang et al., 2019*). The SW region is home to extensive limestone and karst landscapes unsuitable for *C. lanceolata* (*Farooq et al., 2019*). In contrast, the warm and

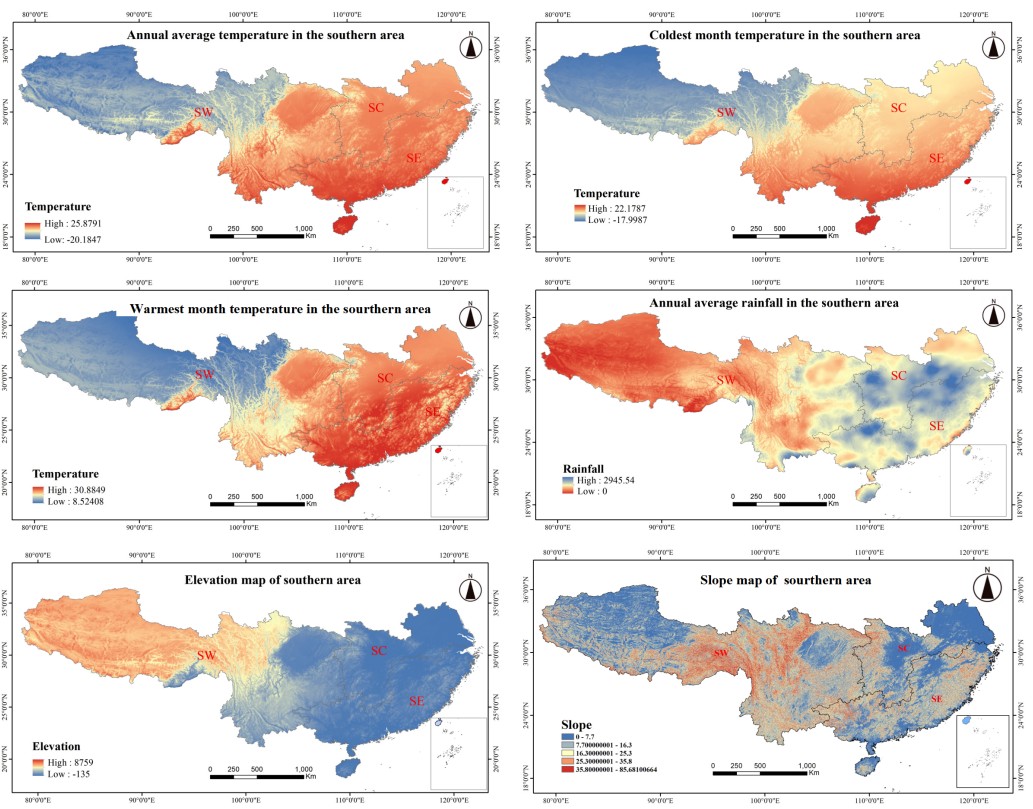

**Figure 4  Climate and topographic factors in southern China.** The various climatic and topographical characteristics across different regions (SW, Southwestern; SC, South-central; SE, Southeastern) in southern China: (A) Annual average temperature distribution. (B) Coldest month temperature distribution. (C) Warmest month temperature distribution. (D) Annual average rainfall distribution. (E) Elevation map. (F) Slope map. Each map highlights the spatial variation in temperature, precipitation, elevation, and slope across the southern regions, which are key factors influencing the growth and distribution of forest species.

humid subtropical maritime and tropical monsoon climates of SE China are ideal for *C. lanceolata*. This species is also highly sensitive to precipitation, with previous work demonstrating that annual average precipitation, driest quarter precipitation, and coldest month temperature are the main environmental factors affecting its distribution (*Chen et al., 2022*; *Feng, 2011*). In contrast, *P. massoniana* prefers well-drained acidic soils and does not tolerate alkaline soils (*Huang et al., 2015*; *Wang & Wang, 2008*; *Meng et al., 2021*). The yellow and yellow-brown soil of the SW is well-suited for *P. massoniana*. Coastal lowlands, however, are alkaline due to seawater intrusion, and are thus unsuitable for *P. massoniana*. Climate and site conditions in SC China are intermediate between SE and SW regions and are relatively suitable for both *P. massoniana* and *C. lanceolata* forests.

There are significant differences in volume growth between age groups for *P. massoniana* and *C. lanceolata*. In SW, both species have consistently higher annual volume growth, especially in the middle-aged stage, compared to SC and SE regions. Studies indicate that summer precipitation has a significant effect on *P. massoniana* during the near-mature

forest stage (*Ni et al., 2023*), although middle-aged trees do not appear to suffer from water stress (*Kang et al., 2017*; *Ettl & Peterson, 1995*; *Zhao et al., 2024*). Conversely, summer heat slows both photosynthesis and growth rates (*Huang, 2021*). Similarly, *C. lanceolata* shows similar growth patterns to *P. massoniana* in middle-aged stands but these patterns diverge following the near-mature stage (*Ali et al., 2019*). Generally, *C. lanceolata* grows better under higher temperatures and rainfall (*Qiao et al., 2022*; *Zhang et al., 2022a*; *Zhang et al., 2022b*), and volume accumulation is higher in places with longer summer growing seasons (*Yin et al., 2021*; *Zhang et al., 2017*). Higher spring temperatures accelerate growth and corresponding volume (*Zhao et al., 2024*; *Wu et al., 2022*; *Chen et al., 2013*). This is consistent with our finding that annual and average volume growth is higher for *C. lanceolata* in SW compared to CS and SE for middle-aged forests. However, growth declines sharply in near-mature forests in SW and ultimately falls below that of SC and SE regions.

This study examined the growth patterns of *C. lanceolata* and *P. massoniana* across southern regions, explored the impact of climate and soil conditions on regional growth differences between the two tree species. SE has ample annual rainfall (>2,000 mm) and fertile acidic soil, both of which are optimal for the rapid growth of *C. lanceolata*. Such warm and humid conditions enhance both wood density and growth rate (*Xie et al., 2024*). In contrast, *P. massoniana* is better adapted to the Southwest's relatively dry, acidic red soil, where the lower phosphorus levels and less fertile soil support root stability and growth under drier conditions (*Hou et al., 2024*; *Guo et al., 2023*). The combined influence of Southeast warmth and Southwest seasonal aridity contributes to the distinct regional adaptations of these species. Therefore, species selection and management strategies tailored to the ecological characteristics of each region are essential for sustainable forest resource development.

We can clearly see that climatic and soil conditions significantly impact the growth of *P. massoniana* and *C. lanceolata*. Therefore, different management approaches should be implemented according to regional climate to optimize the growth of these two species and achieve sustainable management (*Zhang et al., 2022a*; *Zhang et al., 2022b*; *Wang et al., 2023*). In the southwest, favorable climate conditions support rapid growth, but proper management is essential to sustain this growth potential (*Lei et al, 2023*). By contrast, in the southeast, where tree growth is heavily influenced by precipitation, management strategies should focus on maintaining soil moisture and enhancing nutrient supply. The moderate climate of the central-south region makes it suitable for mixed forest models that can enhance both forest productivity and stability (*Cao et al., 2023*). Additionally, soil fertility is a key factor regulating tree growth and varies across regions, thus requiring targeted soil management based on specific soil characteristics. For instance, soils in the southwest may benefit from increased organic matter, while those in the southeast may require more nutrient management.

Research has shown that close-to-nature management strategies can significantly improve soil microbial diversity, thereby enhancing soil fertility and ecosystem stability. Therefore, it is recommended that close-to-nature management be adopted in forest management across all regions to promote biodiversity and ecological stability. To

achieve long-term sustainable forest management, regular monitoring and assessment of forest resources are essential to enable timely adjustments to management strategies, ensuring the rational use and conservation of forest resources while maintaining their ecological functions and biodiversity (*Cao et al., 2023*). Implementing these region-specific management strategies not only supports sustainable forest growth but also balances ecological and economic benefits.

*P. massoniana* and *C. lanceolata* are major afforestation species in southern China, yet they face ecological challenges such as low productivity, efficiency, quality, and poor structural diversity due to both human and natural factors. According to the ninth national forest resource inventory (*Cui & Liu, 2020*), 22.96% of China is forested, covering a total of 220.4462 million hectares and representing 5.51% of the world's forested area. *C. lanceolata* and ponderosa pine are the second and sixth most populous dominant tree species, respectively. However, their average volume per hectare (*C. lanceolata*: 74.83 $m^3/hm^2$; *P. massoniana*: 77.84 $m^3/hm^2$) is below the national average for coniferous forests (94.83 $m^3/hm^2$) due to outdated intensive or extensive forest management practices. Therefore, transitioning these pure coniferous forests into mixed conifer-broadleaf, uneven-aged, near-natural forests through practices like thinning, nurturing, and broadleaf supplementation is recommended to improve multifunctional forest management (*Managi, Wang & Zhang, 2019*; *Lei et al, 2023*). Tailored management strategies based on regional climate and environmental conditions can further ensure sustainable forest management and development (*Landry et al., 2021*).

## CONCLUSIONS

Here, we systematically analyzed large-scale differences in volume growth processes in *P. massoniana* and *C. lanceolata* forests using a unified modeling approach for the first time. This analysis improves understanding of theses trees' growth processes and ecological adaptability. The combined effects of climate and soil conditions is the main reason that southern China is suitable for *P. massoniana* and *C. lanceolata*. In southern China, overall precipitation decreases and site conditions gradually worsen moving inland from coastal areas (*i.e.,* from SE to SW). As a result, the distribution and growth of *C. lanceolata* decline, while those of *P. massoniana* increase. This indicates that SE is more suitable for *C. lanceolata*, whereas SW is more suitable for *P. massoniana*. Both *P. massoniana* and *C. lanceolata* are important and common high-quality, fast-growing coniferous tree species in southern China, with significant ecological, economic, and social value. Therefore, appropriate management approaches should be taken for *P. massoniana* and *C. lanceolata* forests based on different regional and natural environmental climate conditions to accelerate their accumulation growth and structural optimization, achieving sustainable development.

## ACKNOWLEDGEMENTS

We sincerely thank the editor and reviewers for their valuable comments on this manuscript. We acknowledgement the data support from the National Geographic Resource Science

SubCenter, National Earth System Science Data Center, National Science & Technology Infrastructure of China.

### Funding

This study was supported by the following grants: Guizhou Province Forestry Science Project, Grant Number: QLKH[2022]38; National Natural Science Foundation of China, Grant Number: 32001314; Guizhou University Cultivation Project, Grant Number: GDPY[2019]38. The funders had no role in study design, data collection and analysis, decision to publish, or preparation of the manuscript.

### Grant Disclosures

The following grant information was disclosed by the authors:
Guizhou Province Forestry Science Project: QLKH[2022]38.
National Natural Science Foundation of China: 32001314.
Guizhou University Cultivation Project: GDPY[2019]38.

### Competing Interests

The authors declare there are no competing interests.

### Author Contributions

- YuHui Chen conceived and designed the experiments, performed the experiments, analyzed the data, prepared figures and/or tables, authored or reviewed drafts of the article, and approved the final draft.
- Zongzheng Chai conceived and designed the experiments, analyzed the data, authored or reviewed drafts of the article, and approved the final draft.

### Data Availability

The DEM Slope datasets are available at the National Earth System Science Data Center:

– http://dx.doi.org/10.12041/geodata.113786088533256.ver1.db
– http://dx.doi.org/10.12041/geodata.164304785536614.ver1.db
– http://dx.doi.org/10.12041/geodata.65449238360177.ver1.db

The Temperature Precipitation Provincial administrative vector data are available at the Resource and Environmental Science Data Platform:

– http://dx.doi.org/10.12078/2017121301
– http://dx.doi.org/10.12078/2023010103

### Supplemental Information

Supplemental information for this article can be found online at http://dx.doi.org/10.7717/peerj.18706#supplemental-information.

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
