# Peer review of "Comparative analysis of volume growth processes of Masson pine and Chinese fir forests in different regions of southern China"

_PeerJ, doi:10.7717/peerj.18706_

## Round 0.1 · original submission · Major Revisions

In conclusion, this study has significant theoretical and applied ramifications for forest management tactics. To increase the paper's effect, I advise the authors to make a significant rewrite based on the reviewers' recommendations.

Reviewer 1 ·

Basic reporting

1.The English language needs to be retouched, there are some spelling mistakes in the English language, As in the language of Figure3 "he left column represents the growth of Pinus massoniana, while the right column represents Cunninghamia lanceolata. "It should be" The "not" he."
2.The literature reference is sufficient and provides enough background.
3.The article is not marked with section numbers, which makes it difficult for readers to understand the structure.
4.The group of drawings in Figure4 of the article is very fuzzy, and it is difficult to see the numbers clearly, and the compass in the upper right corner of the six drawings is inconsistent with the scale right below. Some of the six drawings are marked SW and SC, while some are not.
5.Richards growth equation is used in the method of data processing in this paper, but the description of the final accuracy verification process is not detailed enough. It is recommended to use R squared and RMSE as the accuracy evaluation results.

Experimental design

no comment

Validity of the findings

1.The paper mentions that there are differences in tree growth between regions, but there is little discussion on this aspect. It can add some differences in precipitation and soil as well as specific effects.
2.According to the growth characteristics and regional differences of Masson pine and Chinese fir, different management methods should be adopted according to climatic conditions in different regions to optimize the growth of these two species, so as to achieve sustainable management. Specific implementation plans can be more specific.

Additional comments

In this paper, the forest growth of masson pine and Chinese fir in different regions of South China was compared and analyzed, focusing on the growth dynamics and ecological adaptability of Masson pine and Chinese fir under different climatic and geographical conditions. Richards growth equation was used to quantify the growth process of trees in different regions, and the growth patterns of two species and their differences in different regions were analyzed. The experimental design of this paper has certain feasibility and innovation, but the accuracy verification analysis of the experimental results has not been fully expressed, and there are also some problems in language writing.

Reviewer 2 ·

Basic reporting

The manuscript clearly expounds the importance of studying the growth dynamics and ecological adaptability of Pinus massoniana and Cunninghamia lanceolata in different regions of southern China. The two tree species are widely distributed in subtropical China and play an important role in the stability of the current and future forest ecosystems in the region and even global climate change. Overall, I believe this paper's rigorous study methodology, thorough data analysis, and insightful conclusion. The purpose of the study is clear, which is of great significance to understand the role of these two important subtropical plantation tree species in the ecosystem, especially in the future carbon neutralization, and to support the sustainable management of plantations.

Experimental design

Using the improved Richards growth equation for large-scale comparative analysis is an innovation. The model performs well in assessing the relationship between volume and age of major tree species in different regions. However, the paper should discuss in more detail why this model was chosen and its comparative advantages with other similar growth models.
The paper mentions the use of national forest resources inventory data and points out the time span of data collection. However, the details of data processing (such as data outlier processing, model verification, etc.) are insufficiently described. It is suggested that more information on these aspects should be added to enhance the credibility of the study.

Validity of the findings

The Results part showed in detail the growth dynamics of the two tree species in different regions, including volume, age-dependent growth pattern and maturity. However, in-depth analysis of the results (such as the ecological explanation of growth differences, the specific effects of climate and soil factors, etc.) is slightly insufficient. It is recommended that discussions in this area be intensified to provide more comprehensive insights.
In the Discussion section, the results of the study were reasonably explained, and the effects of climate and soil conditions on the distribution and growth of tree species were pointed out. However, the discussion of some key findings (such as the reason why the growth of Pinus massoniana is better than that of Cunninghamia lanceolata in Southwest China) is not deep enough. It is suggested that some successful cases or detailed management strategy examples should be added in the discussion section, and the possible mechanisms of differences should be explored to enhance the practicability and guiding significance of the conclusion.

Additional comments

In summary, this is a research paper with important theoretical and practical implications for forest management strategies. I suggest that the authors make a major revision based on the above suggestions to improve the impact of the paper. After revision, I believe this paper will have an important practical guiding significance for the management of coniferous forests such as Pinus massoniana or Cunninghamia lanceolata plantation forest in subtropical China. Other review comments are as follows:

Line 42: “Pinus massoniana (Masson Pine) and Cunninghamia lanceolata (Chinese fir) “can be changed into “Masson Pine (Pinus massoniana) and Chinese fir (Cunninghamia lanceolata) “. And the first appearance of a Latin name in the manuscrpit should contain a named person.
Line 137: the unit “cubic meters per hectare per year” can be abbreviated as” m3·hm-2·yr-1”. Same below.
Line 223: Letter P should be italic.
Line 226: please carefully check “22,044.62 104 hm²”.
The unit of area should be expressed in a unified letter, not in ha or hm2, this can be found in line 229, 230, …, and Table 2 etc..

Annotated reviews are not available for download in order to protect the identity of reviewers who chose to remain anonymous.

Reviewer 3 ·

Basic reporting

See additional comments.

Experimental design

See additional comments.

Validity of the findings

See additional comments.

Additional comments

After careful review, I regret that I cannot accept your paper for publication at this time.

Include a brief background and conclusion in the abstract.
The study has too many aims with no clear focus. Simplifying the research question is recommended.
Clarify how this study differs from published research and its critical significance.
Highlight the most important and unique results, providing an in-depth discussion.
Provide references for any comparative adjectives used.
Summarize your results with generalizations instead of repeating correlations.
Clearly define what is novel in this research that hasn't been previously explored.
Maintain consistent verbal tense throughout the manuscript.
Correct any spacing inconsistencies in sentences.
Improve figure and table captions with more detail.
Ensure correct formatting for items needing superscript on the axes.
Discuss the implications of your findings.
Provide a clear take-home message for readers.
Review all comments and correct any inconsistencies throughout the manuscript.

---

## Round 0.2 · accepted · Accept

This revised manuscript is suitable for publication. Accepted in the current form.

Reviewer 1 ·

Basic reporting

The language quality of the manuscript has significantly improved. The polished text is fluent and accurate, avoiding spelling and grammatical errors, especially the corrected description in Figure 3. It is recommended that the authors maintain this standard in language quality and continue to prioritize linguistic precision in future submissions.
The references are comprehensive and provide sufficient background for the readers. It is suggested to continuously include the latest literature in future studies to maintain the academic depth and timeliness of the research.
The addition of section numbering has significantly improved the structural and logical clarity of the manuscript, enhancing the reading experience. It is recommended that the authors maintain this clear organizational approach in future works.
The revised Figure 4 has significantly improved in terms of clarity, labeling consistency, and detail, enhancing the accuracy and comprehensibility of data presentation. It is recommended that the authors continue to optimize graphical quality in future visualizations to ensure both visual appeal and scientific rigor.

Experimental design

The additions to the accuracy verification section have made the result analysis more comprehensive and transparent. The inclusion of R² and sample details has effectively improved the credibility and reproducibility of the research. It is suggested to continue adopting such precise descriptions in future studies to enhance data reliability.
The revised manuscript has significantly improved in terms of scientific rigor and language fluency. The research design and result presentation are more refined. It is recommended to continue focusing on language editing and methodological rigor in future work.

Validity of the findings

The newly added content clearly explains the impact of precipitation and soil on regional growth patterns, significantly enhancing the scientific and practical value of the study. It is recommended to further explore the effects of more environmental variables in future research to expand ecological understanding.
The newly added management strategies are highly targeted and practical, providing valuable references for sustainable forest management in different regions. It is suggested to validate the effectiveness of these strategies with field case studies in the future.

Additional comments

The authors have provided thorough responses and made appropriate revisions, resulting in a significant improvement in the manuscript's quality. All issues have been effectively addressed. It is recommended that the manuscript be accepted without further revision.

Reviewer 2 ·

Basic reporting

In accordance with my suggestion, the current paper has been revised according to the journal standard template requirements. I am thoroughly satisfied with the overall quality of the article provided. This paper is written in clear, accurate and professional academic language with smooth expression. The terminology used in the article is appropriate and conforms to professional standards in the field of forest management and planning, so that readers can easily understand the main content and ideas of the paper. In the article, the author extensively cited academic achievements in relevant fields and included the latest research papers and previous studies, to provide a solid theoretical foundation and empirical support for the thesis. The article is also equipped with an appropriate amount of charts and tables, including geographical distribution maps, growth curves, etc., which intuitively show the research data and research results, and enhance the persuasiveness and readability of the article.

Experimental design

The authors divided South China into three regions: Southwest, South-central and Southeast China. This method took into account geographical factors, climate and soil conditions, and provided a clear spatial framework for further research. Richards growth equation was selected as the research tool in this paper. The model has wide applicability and accuracy, and can effectively fit the volume growth process of Masson pine and Chinese fir. The author also modified and explained the model according to previous studies to make it more suitable for the needs of this study. The authors obtained climate, terrain and forest resource inventory data from many authoritative data sources to ensure the accuracy and reliability of the data. At the same time, Richards growth equation was applied to fit a large number of plots, which improved the representativeness and universality of the study results.

Validity of the findings

In the current paper, the fitting effect of Richards growth equation is verified, and the results show that the correlation coefficient of most models exceeds 0.6, indicating that the model has high fitting degree and prediction accuracy. This provides a solid data base for subsequent analysis and discussion.

Additional comments

In the discussion or conclusion section of this present paper, it is suggested to add a discussion of the limitations of the study and an outlook for future research directions. Although the findings are discussed in depth in the article, clarifying study limitations helps the reader to understand more fully the scope and limitations of the study. At the same time, the future research direction can provide valuable reference and inspiration for researchers in related fields, and promote the further development of this field.